# Genomic Landscapes of Early-Onset Versus Average-Onset Colorectal Cancer Populations

**DOI:** 10.3390/cancers17050836

**Published:** 2025-02-28

**Authors:** Michael H. Storandt, Qian Shi, Cathy Eng, Christopher Lieu, Thomas George, Melissa C. Stoppler, Elizabeth Mauer, Binyam Yilma, Stamatina Fragkogianni, Emily A. Teslow, Amit Mahipal, Zhaohui Jin

**Affiliations:** 1Department of Oncology, Mayo Clinic, Rochester, MN 55905, USA; storandt.michael@mayo.edu; 2Department of Quantitative Health Sciences, Mayo Clinic, Rochester, MN 55905, USA; 3Division of Hematology and Oncology, Vanderbilt-Ingram Cancer Center, Nashville, TN 37232, USA; 4Division of Medical Oncology, University of Colorado Health Cancer Center, Aurora, CO 80045, USA; 5Division of Hematology and Oncology, University of Florida, Gainesville, FL 32603, USA; 6Tempus AI, Inc., Chicago, IL 60654, USAemily.teslow@tempus.com (E.A.T.); 7Department of Medical Oncology, University Hospitals Seidman Cancer Center, Case Western Reserve University, Cleveland, OH 44106, USA

**Keywords:** early-onset, colorectal cancer, somatic mutation, pathogenic germline variant

## Abstract

Incidence of colorectal cancer among individuals under the age of 50, defined as early-onset colorectal cancer (eoCRC), is increasing. However, the factors driving this are unclear at this time. We assessed the somatic and germline mutational profiles of patients with early-onset and average-onset colorectal cancer (aoCRC, diagnosed at age ≥ 50 years). We found that among 11,006 patients who completed somatic profiling, the most frequently observed somatic mutations included *APC*, *TP53*, and *KRAS*, and the most significant difference between eoCRC and aoCRC was higher rates of *BRAF* mutation among patients with aoCRC. Among 6311 patients who completed germline testing, pathogenic germline variants were discovered in 6.9% of patients with eoCRC and 5.0% of patients with aoCRC. Overall, somatic and germline profiles among those with eoCRC and aoCRC were similar, and do not adequately explain differences in tumor behavior and age of disease onset.

## 1. Introduction

Colorectal cancer (CRC) remains the third most common malignancy globally. Despite a decline in overall incidence, rates of early-onset CRC (eoCRC), defined as cases diagnosed before the age of 50, have increased [1,2]. Within the next 10 years, eoCRC patients are projected to account for up to 10% of new colon cancer and 25% of new rectal cancer diagnoses [3,4,5,6].

Patients with eoCRC are more frequently diagnosed at an advanced stage [7,8], with left-sided tumors and aggressive histopathology [9]. Beyond tumor characteristics, age of diagnosis may have implications for treatment, as recent work has suggested lack of efficacy of *EGFR* inhibition among left-sided tumors in the early-onset population, although the reason for this is currently unknown [10]. Prior studies have shown some differences in the somatic profiles of eoCRC compared to average-onset colorectal cancer (aoCRC, disease diagnosed ≥50 years), including lower prevalence of *BRAF* V600E and *APC* alterations and higher rates of *TP53* alterations [11,12,13]. The implication of differences in the somatic profiles of these cancers remains uncertain at this time.

Additionally, a growing number of studies are investigating the role of germline genetic predisposition in eoCRC. Among all cases of CRC, 2–5% may be associated with a specific cancer syndrome, with one third of cases associated with increased familial risk [14]. Among patients with eoCRC, PGVs have been detected in approximately 12.2–25% of cases [13,15,16,17,18,19,20], with higher rates observed in patients with younger age at time of diagnosis [18,21].

It is critical for us to better understand the role of somatic and germline alterations in the development of eoCRC. Differences in somatic and germline profiles may provide insight into factors that influence predispositions leading to increasing rates of eoCRC. Additionally, by determining the frequency of germline alterations among CRC patients, we can assess the utility of germline screening.

## 2. Methods

### 2.1. Ethics Approval

All patient-level data were deidentified in accordance with the Health Insurance Portability and Accountability Act. Tempus AI, Inc., has been granted an institutional review board exemption (Advarra Pro00072742) permitting the use of deidentified clinical, molecular, and multimodal data to derive or capture results, insights, or discoveries.

### 2.2. Study Design

A cross-sectional analysis was conducted on deidentified records of 11,006 patients with CRC who underwent sequencing between 2017 and 2023. Stages of CRC were included along with sidedness of the primary malignancy if available. Patients were stratified by age of diagnosis (<50, eoCRC vs. ≥50 years, aoCRC). Information collected included patient demographics, tumor immune biomarkers, somatic mutational profiles, and germline alterations.

### 2.3. Molecular Profiling Assay

Next-generation sequencing was conducted using the Tempus xT assay (Tempus AI, Chicago, IL, USA), as previously described [22,23,24,25,26]. Briefly, Tempus xT is a targeted, tumor/normal-matched DNA panel that detects single-nucleotide variants (SNVs), insertions and/or deletions (indels), and copy number variants (CNVs; loss or amplification, copy number 0 or ≥8 respectively) in 596–648 genes, as well as chromosomal rearrangements in 22 genes, with high sensitivity and specificity.

Immune cell infiltration, determined as the proportion of immune cells to tumor and stroma cells, as well as the relative proportions of key immune subtypes, were estimated using gene expression data from xR RNA sequencing. The Tempus immune infiltration algorithm estimates the relative proportion of immune subtypes using a support vector regression (SVR) model, which includes an L2 regularizer and an epsilon insensitive loss function, similar to that of Newman et al. (CIBERSORT) [27]. The SVR was implemented in Python using the nuSVR function in the SVM library of scikit-learn (0.18), with the LM22 reference matrix downloaded from the supplement of Newman et al. [23].

Germline findings from tumor/normal-matched Tempus xT analyses are limited to SNVs and small indels. This germline panel includes 46 hereditary cancer genes (Appendix A). Potential germline findings from Tempus xT sequencing data are reported according to recommendations by the American College of Medical Genetics, National Comprehensive Cancer Network Genetic/Familial High-Risk Assessment guidelines, and other literature [28,29].

### 2.4. Statistical Analysis

Statistical comparisons between eoCRC and aoCRC were conducted using the Wilcoxon rank sum test and the Pearson’s Chi-squared test for continuous variables and categorical variables, respectively. Analyses were two-sided and statistical significance was established at an alpha level of 0.05, indicating that differences with a *p*-value below this threshold were considered significant. All analyses were performed in R (version 4.2.3.).

## 3. Results

### 3.1. Cohort Characteristics

A total of 11,006 patients were included in the overall cohort, including 2379 eoCRC patients and 8627 aoCRC patients. Within the overall cohort, 57% were male, 76% were white, and 80% had stage 4 disease (Table 1). Among patients with eoCRC, 85% had a left sided/rectal primary, whereas only 75% with aoCRC had a left-sided/rectal primary (*p* < 0.001, Table 1).

### 3.2. Immunological Markers

Among patients with eoCRC and aoCRC, tumors were found to have high tumor mutation burden (≥10 mutations/megabase, TMB-H) in 5.7% and 7.6%, respectively (*p* = 0.002), and were MSI-H/dMMR in 4.2% and 6.8%, respectively (*p* < 0.001). Positive programmed death-ligand 1 (PD-L1) expression was seen in 3.4% and 4.9% of patients with eoCRC and aoCRC, respectively (*p* = 0.071) (Table 2). Among patients who completed RNA sequencing, immune cell infiltration was comparable between eoCRC and aoCRC patients (Appendix A).

### 3.3. Somatic Mutation Profiles

Next, evaluating somatic mutation profiles of patients with eoCRC and aoCRC, respectively, the most commonly observed somatic mutations included *TP53* in 68.7% and 65.9%, *APC* in 65.9% and 68.6%, *KRAS* in 43% and 43%, *SMAD4* in 17.2% and 14.8%, *PIK3CA* in 14.5% and 15.7%, *ALK* in 14.4% and 14.4%, and *RET* in 12.2% and 12%. A full list of somatic alterations seen in >3% of patients is shown in Figure 1a, Appendix A. The most common somatic alterations when looking only at MSS/pMMR tumors in patients with eoCRC and aoCRC, respectively, included *APC* (67.2% and 72.6%), *TP53* (71.5% and 69.9%), *KRAS* (43.6% and 45.2%), *SMAD4* (17.8% and 15.5%), *PIK3CA* (13.5% and 15%), *ALK* (14.2% and 14.1%), and *RET* (12.3% and 11.7%) (Figure 1b). A co-occurrence analysis, as depicted in Appendix A, did not show any notable differences in frequency of somatic variant co-occurrence between those with eoCRC and aoCRC.

When listing somatic alterations by order of most significant *q*-value between patients with eoCRC and aoCRC, most significant differences, with prevalence higher among those with aoCRC, were noted in *BRAF* (4.7 vs. 9.8%, *q* < 0.001), *RNF43*, (2.9% vs. 6.0%, *q* < 0.0001), and *AMER1* (3.5% vs. 5.8%, *q* < 0.0001), respectively. Among genes more commonly altered in eoCRC patients compared to aoCRC patients, most significant differences were noted in *UGT2B28* (1.7% vs. 0.8%, *q* = 0.09), *NKX2-1* (3.5% vs. 2.3%, *q* = 0.022), and *C15orf40* (2.6% vs. 1.6%, *q* = 0.022), respectively. A list of somatic alterations with *q*-value < 0.05 are listed in Table 3 (full list of somatic alterations is available in Appendix A).

As mentioned, compared to eoCRC, *BRAF* was more frequently altered in patients with aoCRC (4.7% vs. 9.8%, *p* < 0.0001). The most common *BRAF* mutation among patients with eoCRC and aoCRC was V600E, mutated in 3.0% and 7.7%, respectively. Non-V600E *BRAF* mutations were present in 1.7% of eoCRC patients and 2.1% of aoCRC patients. A full list of *BRAF* mutations is listed in Appendix A. Among patients with MSI-H/dMMR tumors with TMB-H, 1.2% with eoCRC and 49.2% with aoCRC had a *BRAF* V600E mutation (*p* < 0.001). Among patients with MSS/MSI-L tumors with low TMB, 3.5% with eoCRC and 5.5% with aoCRC had a V600E mutation (*p* < 0.0001). Frequency of *KRAS* alteration was not significantly different between eoCRC and aoCRC, seen in 43% in each group. *KRAS* mutations are listed in Appendix A.

Further analysis was performed to assess for frequency of CNVs (both amplifications and deletions) among eoCRC and aoCRC patients. CNVs are listed in order of significance of difference between eoCRC and aoCRC patients in Appendix A. The most significant differences between eoCRC and aoCRC patients, respectively, included CNVs involving *ERBB2* (3.7% vs. 2.3%, *q* = 0.015), *NKX2-1* (3.5% vs. 2.3%, *q* = 0.037), and *CDK12* (2.3% vs. 1.3%, *q* = 0.037). There was no difference in frequency of *MYC* amplification between eoCRC and aoCRC patients (1.9% vs. 1.6%, *q* = 0.6).

Four hundred and ninety-two eoCRC (20.7%) and 1411 aoCRC (16.4%) patients had left-sided primaries with wild-type *BRAF*, *NRAS*, and *KRAS.* Given the recently reported observation that eoCRC patients with *KRAS* wild-type left-sided primary tumors may not benefit from first line EGFR inhibition, we explored the potential role of negative hyperselection (defined as the presence of at least one pathogenic or likely pathogenic point mutation in any of the following genes within a tumor: *KRAS*, *NRAS*, *BRAF* (only V600E), *AKT1*, *ERBB2*, *PIK3CA* (exons 9 and 20), *PTEN*, *ALK1*, or *ERBB2* amplification). Conversely, the hyperselected wild-type subgroup included patients not exhibiting any of these mutations [30]. In the present study, the eoCRC population had a higher frequency of *ERBB2* amplification (4.8% vs. 3.7%, *q* = 0.030), but a lower prevalence of *BRAF* V600E mutation (3.0% vs. 7.7%, *q* < 0.001) (Table 4). The prevalence of other mutations was similar in both groups and could not explain the observed treatment efficacy differences.

### 3.4. Germline Profiling

In total, 1413 (59.4%) with eoCRC and 4898 (56.8%) with aoCRC underwent germline testing in conjunction with somatic mutation profiling. Among these, 6.9% with eoCRC and 5.0% with aoCRC were found to have a PGV (*p* = 0.006). The most commonly observed PGVs in patients with eoCRC and aoCRC included *MUTYH* (1.3% and 1.7%), *ATM* (0.8% and 0.4%), *APC* (0.6% and 0.2%), *CHEK2* (0.5% and 0.4%), *BRCA2* (0.4% and 0.3%), and *MSH2* (0.4% and 0.2%), respectively (Figure 2, Appendix A). PGVs with the largest difference in frequency between patients with eoCRC and aoCRC included *TP53* (0.4% vs. <0.1%, *q* = 0.2), *APC* (0.6% vs. 0.2%, *q* = 0.4), *ATM* (0.8% vs. 0.4%, *q* = 0.4), and *RAD51C* (0.3% vs. <0.1%, *q* = 0.4), respectively (Appendix A).

When assessing the frequency of high penetrance PGVs in patients with eoCRC and aoCRC, the most common PGVs included *MUTYH* in 1.3% and 1.7% (*q* > 0.9), *APC* in 0.6% and 0.2% (*q* = 0.3), *BRCA2* in 0.4% and 0.3% (*q* > 0.9), *MSH2* in 0.4% and 0.2% (*q* > 0.9), and *MLH1* in 0.3% and 0.2% (*q* > 0.9), respectively (Appendix A). The most commonly observed germline mutations in eoCRC and aoCRC patients with MSI-L/MSS tumors included *MUTYH* in 1.4% and 1.8% (*p* = 0.3), *ATM* in 0.8% and 0.4% (*p* = 0.053), *APC* in 0.7% and 0.2% (*p* = 0.029), and *CHEK2* in 0.5% and 0.4% (*p* = 0.5), respectively (Appendix A). The most common germline mutations in eoCRC and aoCRC among patients with MSI-H/dMMR tumors included *MSH2* in 9.8% and 1.7% (*p* = 0.008), *MLH1* in 7.8% and 3.7% (*p* = 0.3), *PMS2* in 5.9% and 2% (*p* = 0.13), and *MSH6* in 2% and 3% (*p* > 0.9), respectively (Appendix A).

### 3.5. Incidental Finding of a Pathogenic/Likely Pathogenic Germline Variant in a Lynch Syndrome Gene

As previously mentioned, Tempus xT reports germline incidental findings on a limited set of variants associated with inherited cancer syndromes. We conducted subsequent analysis of 14 eoCRC and 45 aoCRC patients with incidental detection of a pathogenic/likely pathogenic variant in a Lynch syndrome gene. Germline mutations seen in those with eoCRC included *MSH2* (five patients), *MLH1* (four patients), *PMS2* (three patients), and *MSH6* (two patients). PGVs found in aoCRC patients included *MSH6* (13 patients), *MLH1* and *PMS2* (12 patients each), and *MSH2* (8 patients). Overall, the somatic landscape was similar between patients with eoCRC and aoCRC who had an incidentally detected PGV in a Lynch syndrome gene (Appendix A). Patients with eoCRC with incidental detection of a PGV in a Lynch syndrome gene were more likely to have a TMB-H tumor when compared to aoCRC patients (93% vs. 69%, *p* = 0.090), although this was not statistically significant. Tumor mutation burden in patients with eoCRC and aoCRC with a germline mutation in a Lynch syndrome gene is depicted in Appendix A. Similarly, among eoCRC patients with a PGV in a Lynch syndrome gene, 93% had an MSI-H/dMMR tumor while only 69% of aoCRC patients had an MSI-H/dMMR tumor (*p* = 0.09).

We then looked at 22 eoCRC patients and 130 aoCRC patients with MSI-H/dMMR tumors who were not found to have a PGV in a Lynch syndrome gene. Of these patients, no PGVs were found in those with eoCRC, while a PGV was found in five aoCRC patients, including *ATM*, *BRIP1*, *CHEK2*, *MSH3*, and *PALB2* (one patient each).

## 4. Discussion

In this large, cross-sectional study, we sought to evaluate the somatic and germline mutation profiles of patients with eoCRC and compare them to patients with aoCRC to determine if a unique profile may explain discrepancies in tumor characteristics and provide clues as to the etiology of the recent increase in rates of eoCRC. Additionally, by quantifying frequency of PGVs in patients with CRC, we were able to evaluate the value of universal germline screening for all CRC patients. To our knowledge, this is one of the largest studies to do so and provides valuable information to guide management of CRC in these populations.

First, looking at the somatic profiles of those with eoCRC and aoCRC, some differences were noted. Among pathogenic mutations with implications for clinical management in those with eoCRC and aoCRC, *BRAF* mutations were more common among patients with aoCRC (9.8% vs. 4.7%), which is similar to prior studies that have reported increased frequency of *BRAF* mutations in the average-onset population [11,12,13,31]. Importantly, our demonstration of increased *BRAF* mutations in the aoCRC population models what was seen under universal testing conditions, further adding strength to the generalizability of these results being less impacted by selection bias [32]. Overall, however, the most frequently observed somatic mutations among those with eoCRC and aoCRC were strikingly similar, which has been noted in a prior study [33]. Regardless, this does not discount the importance of somatic mutational profiling for patients with CRC, as actionable mutations, such as those involving *KRAS*, are detected in a large proportion of patients. In the present study, frequency of dMMR/MSI-H was lower in those with eoCRC compared to aoCRC, which does conflict with prior studies [11,15,16]. Additionally, the absolute number of dMMR/MSI-H tumors in the whole cohort was lower in this study. This may be explained by the fact that 80% of patients included in this study had stage 4 disease, and prior studies have shown that rates of Lynch syndrome detection decrease as disease stage advances [34,35]. We do, however, demonstrate that among aoCRC patients with MSI-H/dMMR tumors, nearly half were in the setting of *BRAF* V600E, suggesting a sporadic mutation, while only 1% of those with eoCRC were found to have a *BRAF* V600E mutation in this setting. Further, among patients with dMMR/MSI-H tumors, a PGV in a Lynch syndrome gene was incidentally detected in 25% of eoCRC patients and only 10% of aoCRC patients. This would suggest that MSI-H/dMMR is less frequently sporadic in eoCRC populations. When looking at patients with MSI-L/MSS tumors, aoCRC patients still had a higher frequency of *BRAF* mutation.

Next, evaluating the detection of PGVs in patients with eoCRC, the reported frequency of 6.9% is lower than prior studies reporting a frequency of 14–25% [15,16,17,18,19,20]. Multiple explanations exist as to why detection frequencies were lower in the presented study, but one possibility is that this number is more reflective of the prevalence of PGVs in stage IV CRC patients. This study utilized a large database of patients who underwent somatic profiling of their tumor to guide therapeutic options, and many (57.3%) elected to obtain matched normal sequencing to improve the accuracy of detection of somatic mutations, MSI, and TMB, as well as potential germline findings. We suspect that this population is less selected than prior studies where patients may have had to agree to complete a separate test for germline sequencing that was not coupled with somatic mutation profiling or to see a genetic counselor. Alternatively, this study may underestimate true prevalence, as this analysis included a majority of patients with metastatic disease, which may impact frequency of PGVs. For example, Lynch syndrome is the most common etiology of hereditary CRC, and the prevalence of Lynch syndrome has been shown to decrease as disease stage advances [34,35]. The low prevalence of Lynch syndrome mutations detected in our study (1.0% in our study vs. 5.1% reported by prior meta-analysis) is also consistent with this hypothesis [36]. In addition, the panel does have limitations in terms of the germline mutations that might be captured in patients who completed matched normal sequencing, as it has not been validated for assessment of hereditary disease risk and thus the number of false positives and false negatives has not been determined and certain alterations are not able to be detected (e.g., single- and multi-exon level deletions/duplications and gene rearrangements). Additionally, detection of genetic variation in genes with high homology to other regions of the genome may be decreased or not reliably detected by NGS (including *PMS2*). Mosaic germline alterations are also not detectable by this assay. Because of these limitations, these germline test results cannot be used to definitively rule out genetic predisposition syndromes, such as Lynch syndrome.

Despite lower detection of PGVs, this study does continue to reinforce the importance of germline screening for all patients with CRC, considering PGV detection in 5% with aoCRC and 6.9% with eoCRC. Current National Comprehensive Cancer Network (NCCN) guidelines recommend germline screening for all patients with eoCRC, and suggests additional risk-based screening recommendations for any individual with a ≥5% risk of having a PGV in an MMR gene based on predictive models, such as PREMM5, MMRpro, or MMRpredict [37]. In the case of PREMM5, all male CRC patients diagnosed at age < 65 should undergo germline screening; however, this is not routinely conducted in clinical practice [38].

In recent history, there has been growing support for universal germline testing among individuals with cancer [39]. However, there are issues that arise with universal germline screening. For example, in the above study, the most frequently observed mutation among those with eoCRC and aoCRC was *MUTYH*. Biallelic *MUTYH* mutations may result in *MUTYH*-associated adenomatous polyposis; however, monoallelic mutation has been found to confer no increased risk of CRC [40], and as such, NCCN guidelines recommend standard risk screening for CRC in this population [37]. In fact, many of the PGVs detected in this study and others are not known to increase risk of CRC, and in many cases, such as monoallelic *MUTYH*, have no known clinical significance. Detection of these mutations, therefore, has the potential to result in increased distress on behalf of the patient, without clinical benefit.

Lastly, when evaluating the germline profiles of those with eoCRC and aoCRC, there is significant similarity. It is difficult to entirely explain the difference in age of development of cancer by looking at PGVs in isolation. Future studies may aim to evaluate polygenic risk, similar to what has been studied in prostate cancer [41]. Efforts are currently ongoing to develop polygenic risk scores in CRC and may be a means to distinguish between those who will develop CRC prior to recommended screening initiation and those who will develop CRC later in life [42,43]. Additionally, understanding the impact of lifestyle and diet is also of importance.

Limitations of this study include high predominance of white, non-Hispanic patients with metastatic disease, which may influence prevalence of PGVs when compared to prior studies [44]. Additionally, approximately twice as many patients underwent somatic tumor only profiling compared to those who also completed matched normal sequencing, and it is challenging to assess whether influencing factors may have had an impact on which patients decided to pursue germline profiling. This real-world dataset is also impacted by selection bias and is relevant to only those patients for whom commercial genetic profiling was successfully obtained. However, our results are consistent in many regards to published reports of universal somatic mutation profiling, suggesting the impact of selection bias is waning as more patients are having tumor profiling performed.

## 5. Conclusions

In this study, we show that while differences exist in the somatic profiles of eoCRC and aoCRC, the frequency of various somatic alterations is quite similar, and may not adequately account for differences in tumor behavior. Additionally, we report PGV detection in 6.9% of patients with eoCRC and 5.0% with aoCRC, which is lower than prior studies and may represent a better reflection of true frequency of PGVs in patients with stage IV CRC, although limitations with this assay exist. Despite lower incidence of PGVs in this population compared to prior reports, this incidence of detection would still suggest that universal germline screening may be indicated for all patients with CRC.

## Figures and Tables

**Figure 1 cancers-17-00836-f001:**
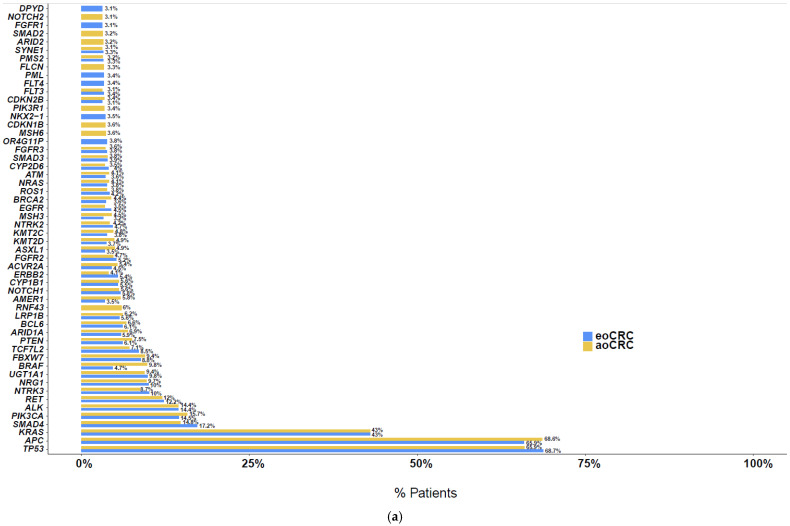
Somatic alterations in >3% of (**a**) all patients and (**b**) patients with MSS/pMMR tumors.

**Figure 2 cancers-17-00836-f002:**
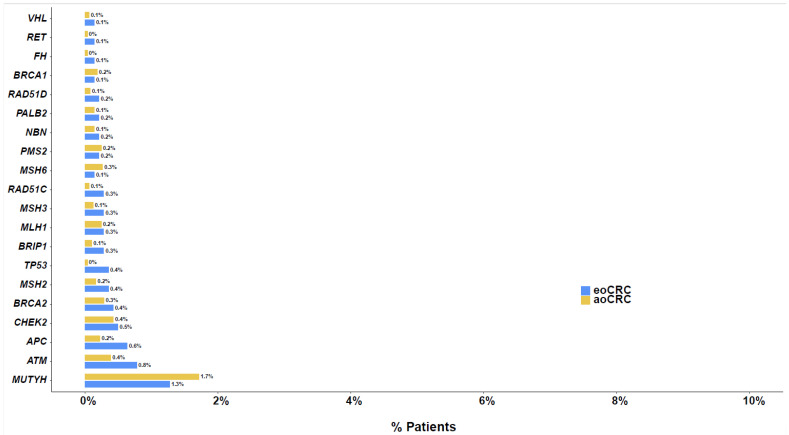
Germline mutations in >0.1% of patients with eoCRC and aoCRC.

**Table 1 cancers-17-00836-t001:** Cohort characteristics of patients with early-onset (eoCRC) and average-onset (aoCRC) colorectal cancer.

	Overall, N = 11,006	eoCRC, N = 2379	aoCRC, N = 8627	*p*-Value ^b^
Age at Diagnosis, median (IQR)	60 (51, 69)	43 (38, 47)	64 (57, 72)	<0.001
Gender				<0.001
Male	6242 (57%)	1275 (54%)	4967 (58%)	
Female	4764 (43%)	1104 (46%)	3660 (42%)	
Race				<0.001
White	4923 (76%)	931 (72%)	3992 (77%)	
Black or African American	853 (13%)	187 (14%)	666 (13%)	
Other	425 (6.6%)	109 (8.4%)	316 (6.1%)	
Asian	270 (4.2%)	67 (5.2%)	203 (3.9%)	
Unknown	4535	1085	3450	
Ethnicity				<0.001
Not Hispanic or Latino	3165 (83%)	600 (76%)	2565 (85%)	
Hispanic or Latino	660 (17%)	194 (24%)	466 (15%)	
Unknown	7181	1585	5596	
Stage ^a^				0.03
Stage 4	6667 (80%)	1389 (81%)	5278 (80%)	
Stage 3	1162 (14%)	245 (14%)	917 (14%)	
Stage 2	424 (5.1%)	63 (3.7%)	361 (5.5%)	
Stage 1	65 (0.8%)	14 (0.8%)	51 (0.8%)	
Unknown	2688	668	2020	
Laterality (only pts with colorectal tissue sequenced)				<0.001
Left	1903 (43%)	492 (50%)	1411 (42%)	
Rectum	1467 (33%)	348 (35%)	1119 (33%)	
Right	801 (18%)	117 (12%)	684 (20%)	
Transverse colon	214 (4.9%)	36 (3.6%)	178 (5.2%)	
Unknown	6621	1386	5235	
Time from diagnosis to sample collection				<0.001
≤3 months	7639 (69%)	1558 (65%)	6081 (70%)	
>3 to ≤6 months	379 (3.4%)	99 (4.2%)	280 (3.2%)	
>6 to ≤12 months	642 (5.8%)	172 (7.2%)	470 (5.4%)	
>1 year to ≤2 years	1009 (9.2%)	232 (9.8%)	777 (9.0%)	
>2 years to 5 years	1337 (12%)	318 (13%)	1019 (12%)	

^a^ Within 60 days of sample collection; ^b^ Wilcox rank sum test, Pearson’s Chi-squared test; Abbreviations: aoCRC: average-onset colorectal cancer, eoCRC: early onset colorectal cancer.

**Table 2 cancers-17-00836-t002:** Tumor immunological markers in patients with early-onset (eoCRC) and average-onset (aoCRC).

Characteristic	Overall, N = 11,006	eoCRC, N = 2379	aoCRC, N = 8627	*p*-Value ^a^
TMB				0.002
<10	9471 (93%)	2054 (94%)	7417 (92%)	
≥10	738 (7.2%)	124 (5.7%)	614 (7.6%)	
Unknown	797	201	596	
MSI-H/dMMR				<0.001
MSI Low/Stable	10,119 (94%)	2235 (96%)	7884 (93%)	
MSI-H and/or dMMR	672 (6.2%)	97 (4.2%)	575 (6.8%)	
Unknown	215	47	168	
PD-L1 result from internal IHC				0.071
Negative	3760 (95%)	819 (97%)	2941 (95%)	
Positive	180 (4.6%)	29 (3.4%)	151 (4.9%)	
Unknown	7066	1531	5535	
MSI/MMR/TMB Status				
MSI Low/Stable, TMB < 10	9339 (85%)	2030 (85%)	7309 (85%)	
MSI Low/Stable	680 (6.2%)	171 (7.2%)	509 (5.9%)	
MSI-H and/or dMMR, TMB ≥ 10	624 (5.7%)	87 (3.7%)	537 (6.2%)	
MSI Unknown	215 (2.0%)	47 (2.0%)	168 (1.9%)	
MSI Low/Stable, TMB ≥ 10	100 (0.9%)	34 (1.4%)	66 (0.8%)	
MSI-H and/or dMMR	35 (0.3%)	9 (0.4%)	26 (0.3%)	
MSI-H and/or dMMR, TMB < 10	13 (0.1%)	1 (<0.1%)	12 (0.1%)	

^a^ Wilcox rank sum test, Pearson’s Chi-squared test; Abbreviations: aoCRC: average-onset colorectal cancer, dMMR: deficient mismatch repair, eoCRC: early onset colorectal cancer, MSI-H: high microsatellite instability, PD-L1: programmed death-ligand 1, TMB: tumor mutation burden.

**Table 3 cancers-17-00836-t003:** Somatic alterations in patients with eoCRC and aoCRCby significance of *q*-value.

	eoCRC, N = 2379 ^a^	aoCRC, N = 8627 ^a^	*p*-Value ^b^	*q*-Value ^c^
*BRAF*	111 (4.7%)	845 (9.8%)	<0.001	<0.001
*RNF43*	68 (2.9%)	515 (6.0%)	<0.001	<0.001
*AMER1*	83 (3.5%)	503 (5.8%)	<0.001	<0.001
*ZNRF3*	24 (1.0%)	192 (2.2%)	<0.001	0.008
*UGT2B28*	40 (1.7%)	71 (0.8%)	<0.001	0.009
*RPL22*	24 (1.0%)	180 (2.1%)	<0.001	0.02
*NKX2-1*	84 (3.5%)	199 (2.3%)	<0.001	0.022
*C15orf40*	63 (2.6%)	139 (1.6%)	<0.001	0.022
*MSH6*	54 (2.3%)	312 (3.6%)	0.001	0.024
*GNAS*	27 (1.1%)	187 (2.2%)	0.001	0.024
*HNF1A*	23 (1.0%)	168 (1.9%)	0.001	0.024
*FLCN*	50 (2.1%)	289 (3.3%)	0.002	0.027
*SEC31A*	39 (1.6%)	77 (0.9%)	0.002	0.027
*FAS*	14 (0.6%)	119 (1.4%)	0.002	0.027
*ASXL1*	83 (3.5%)	426 (4.9%)	0.003	0.036
*CDK12*	70 (2.9%)	167 (1.9%)	0.003	0.036
*PRSS1*	30 (1.3%)	56 (0.6%)	0.003	0.036
*SMAD4*	409 (17%)	1274 (15%)	0.004	0.043
*ERBB2*	129 (5.4%)	350 (4.1%)	0.004	0.043
*SMARCE1*	24 (1.0%)	43 (0.5%)	0.005	0.049

^a^ n (%); ^b^ Pearson’s Chi-squared test; ^c^ False discovery rate correction for multiple testing; Abbreviations: aoCRC: average-onset colorectal cancer, eoCRC: early onset colorectal cancer.

**Table 4 cancers-17-00836-t004:** Frequency of somatic alterations among hyperselection genes in patients with eoCRC and aoCRC.

Characteristic	eoCRC, N = 2379 ^a^	aoCRC, N = 8627 ^a^	*p*-Value ^b^	*q*-Value ^c^
*BRAF*	71 (3.0%)	664 (7.7%)	<0.001	<0.001
*ERBB2*	114 (4.8%)	316 (3.7%)	0.012	0.030
*PTEN*	98 (4.1%)	392 (4.5%)	0.4	0.6
*NRAS*	85 (3.6%)	335 (3.9%)	0.5	0.6
*KRAS*	1002 (42%)	3647 (42%)	0.9	0.9

^a^ n (%). ^b^ Pearson’s Chi-squared test. ^c^ False discovery rate correction for multiple testing.

## Data Availability

Data used in the research was collected in a real-world health care setting and is subject to controlled access for privacy and proprietary reasons. When possible, derived data supporting the findings of this study have been made available within the paper and its Appendix A. Tempus may make access to further de-identified data available pending a signed data use agreement, with data provisioned via Tempus LENS. Inquiries should be addressed to LENS@tempus.com.

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
