# Peer review of "Genomic Landscapes of Early-Onset Versus Average-Onset Colorectal Cancer Populations"

_cancers, 2025, doi:10.3390/cancers17050836_

Round 1
Reviewer 1 Report
Comments and Suggestions for Authors
Storandt and colleagues present a detailed comparison of somatic and germline mutations on a large cohort of early-onset and average-onset colorectal cancer patients (eoCRC and aoCRC respectively). They found that the most significant difference between eoCRC and aoCRC was higher rates of BRAF mutations in aoCRC patients. Pathogenic germline variants were detected in 6.9% and 5% of eoCRC and aoCRC patients respectively, lower rates than previously reported. Overall, the study is presented clearly, rigorous statistical testing is applied, and the limitations of the cohort are carefully considered.
However, the somatic findings mostly agree with previous literature, so the novelty of the study relies on the size of the cohort. Additionally, the authors’ admit that the lower detection of pathogenic germline variants could be confounded by cohort and assay biases. Nevertheless, the study is of sufficient quality and significance for publication, once specific comments below have been addressed:
Major comments
1) The authors state that the Tempus xT assay includes copy number alterations (CNAs), but this data is only mentioned once in the context of ERBB2 amplification (line 172). Was systematic analysis carried out on differences in gene amplification/deletion between eoCRC and aoCRC (for example, there is conflicting evidence that MYC amplification is more common is eoCRC: https://pmc.ncbi.nlm.nih.gov/articles/PMC9496520/ )? Or even mutual exclusivity/cooccurrence of mutation and CNA of the main driver genes? A paper entitled “Genomic landscapes…” should consider CNAs when the data is available, especially in colorectal cancer where CNAs are a defining characteristic between MSS and MSI tumours.
2) RNA sequencing data is used for immune cell infiltration estimates (line 125), yet there is no information on the wet/dry lab processing of the RNA-sequencing that led to the immune cell infiltration data. I appreciate that the authors were likely not the primary collectors of the data, but there needs to be a paragraph on RNA-sequencing in the methods (i.e. in 2.3. Molecular Profiling Assay), akin to the explanation of the Tempus xT assay.
Minor comments
1) For consistency between the main text and the figures (including supplementary), all legend labels of “<50” and “>=50” should be replaced with “eoCRC” and “aoCRC” respectively. These legends should also be larger.
2) Line 105-106 – Were all statistical tests two-sided? This should be stated explicitly.
3) Figure 1 – As a suggestion, the figure (both panels) might be easier to interpret if the % difference between eoCRC and aoCRC is plotted instead of the % for each cohort. The order could be kept the same, so it is obvious that the genes at the bottom had the highest prevalence.
4) Table 3 – The authors should comment on the directionality of the differences. For instance, UGT2B28 is the gene with the most significant difference between eoCRC and aoCRC in which a higher percentage of mutants is found in eoCRC.
5) Line 155 – It would aid readability if percentages were always reported the same way around (i.e. eoCRC before aoCRC), especially in the same paragraph. For instance, the line 155 sentence should read: “The most common BRAF mutation among patients with eoCRC and 154 aoCRC was V600E, mutated in 3.0% and 7.7%, respectively”
6) Line 155-158 – The large difference between MSI eoCRC and aoCRC for BRAF V600E is surprising, along with the fact that a greater percentage of MSS eoCRCs have BRAF V600E mutations compared to MSI eoCRCs (all preceding literature finds V600E to be more common in MSI vs MSS, e.g. https://pmc.ncbi.nlm.nih.gov/articles/PMC4580394/). Could the authors please check these numbers are correct, particularly the 1.2% in eoCRC MSI-H?
7) Line 181 – Is the difference in PGV % between eoCRC and aoCRC significant? It would be good to show the p-value of this comparison (or just state it wasn’t significant if it wasn’t).
8) Line 193-194 and 195-197 – Please report the significance value of these comparisons between eoCRC and aoCRC. This could be in the text, annotated on Supplemental 13, or added as an extra Supplemental table.
9) Figure 2 – The axis label “% Patients” should be on the x-axis instead of the y-axis
10) Line 215/216 – What is the significance value of this comparison?
11) Line 248-251 – The argument about MSI being less frequently sporadic should be explicitly assessed by comparing the percentage of eoCRC vs. aoCRC patients that have confirmed Lynch syndrome in the cohort.
12) Line 323-324 – The final line of the conclusion seems slightly incongruous with the results; stating that “Based on this frequency of detection, universal germline screening may be considered in all patients with CRC” when this study found lower frequency than previous studies is slightly mis-leading. Please consider removing or modifying this statement.
Author Response
Major comments
1) The authors state that the Tempus xT assay includes copy number alterations (CNAs), but this data is only mentioned once in the context of ERBB2 amplification (line 172). Was systematic analysis carried out on differences in gene amplification/deletion between eoCRC and aoCRC (for example, there is conflicting evidence that MYC amplification is more common is eoCRC: https://pmc.ncbi.nlm.nih.gov/articles/PMC9496520/ )? Or even mutual exclusivity/cooccurrence of mutation and CNA of the main driver genes? A paper entitled “Genomic landscapes…” should consider CNAs when the data is available, especially in colorectal cancer where CNAs are a defining characteristic between MSS and MSI tumors.
Response 1: We have now added comparative data for CNVs (both amplifications and deletions), between patients with eoCRC and aoCRC. In our dataset we determined that MYC amplification, which occurred in 1.9% (n=46) of eoCRC and 1.6% (n=139) of aoCRC patients, was not significantly different (p=0.6).
2) RNA sequencing data is used for immune cell infiltration estimates (line 125), yet there is no information on the wet/dry lab processing of the RNA-sequencing that led to the immune cell infiltration data. I appreciate that the authors were likely not the primary collectors of the data, but there needs to be a paragraph on RNA-sequencing in the methods (i.e. in 2.3. Molecular Profiling Assay), akin to the explanation of the Tempus xT assay.
Response 2: We have now added a description regarding RNA-sequencing methods used for determination of immune cell infiltration in the methods section.
Minor comments
1) For consistency between the main text and the figures (including supplementary), all legend labels of “<50” and “>=50” should be replaced with “eoCRC” and “aoCRC” respectively. These legends should also be larger.
Response 1: Figures in the manuscript have now been updated with the recommended changes.
2) Line 105-106 – Were all statistical tests two-sided? This should be stated explicitly.
Response 2: We now specify that all analyses were two-sided.
3) Figure 1 – As a suggestion, the figure (both panels) might be easier to interpret if the % difference between eoCRC and aoCRC is plotted instead of the % for each cohort. The order could be kept the same, so it is obvious that the genes at the bottom had the highest prevalence.
Response 3: We appreciate this suggestion, but feel Figure 1 may be best if left as is to show relative frequency of the most common mutations across the board.
4) Table 3 – The authors should comment on the directionality of the differences. For instance, UGT2B28 is the gene with the most significant difference between eoCRC and aoCRC in which a higher percentage of mutants is found in eoCRC.
Response 4: We now indicate directionality, and make specific mention of most significant differences, among genes more frequently altered among eoCRC patients in the body of the manuscript.
5) Line 155 – It would aid readability if percentages were always reported the same way around (i.e. eoCRC before aoCRC), especially in the same paragraph. For instance, the line 155 sentence should read: “The most common BRAF mutation among patients with eoCRC and 154 aoCRC was V600E, mutated in 3.0% and 7.7%, respectively”
Response 5: We have carefully reviewed the results section, and now always list eoCRC prevalence first, so that there is symmetry throughout for ease of reading.
6) Line 155-158 – The large difference between MSI eoCRC and aoCRC for BRAF V600E is surprising, along with the fact that a greater percentage of MSS eoCRCs have BRAF V600E mutations compared to MSI eoCRCs (all preceding literature finds V600E to be more common in MSI vs MSS, e.g. https://pmc.ncbi.nlm.nih.gov/articles/PMC4580394/). Could the authors please check these numbers are correct, particularly the 1.2% in eoCRC MSI-H?
Response 6: We have reviewed, and these numbers are correct.
7) Line 181 – Is the difference in PGV % between eoCRC and aoCRC significant? It would be good to show the p-value of this comparison (or just state it wasn’t significant if it wasn’t).
Response 7: We have now included this p-value, which does show statistical significance.
8) Line 193-194 and 195-197 – Please report the significance value of these comparisons between eoCRC and aoCRC. This could be in the text, annotated on Supplemental 13, or added as an extra Supplemental table.
Response 8: We have now included these p-values in the body of the manuscript.
9) Figure 2 – The axis label “% Patients” should be on the x-axis instead of the y-axis
Response 9: This is now corrected.
10) Line 215/216 – What is the significance value of this comparison?
Response 10: This p-value is now included.
11) Line 248-251 – The argument about MSI being less frequently sporadic should be explicitly assessed by comparing the percentage of eoCRC vs. aoCRC patients that have confirmed Lynch syndrome in the cohort.
Response 11: We now state in the discussion, “Further, among patients with dMMR/MSI-H tumors, a PGV in a Lynch syndrome gene was incidentally detected in 25% of eoCRC patients and only 10% of aoCRC patients.”
12) Line 323-324 – The final line of the conclusion seems slightly incongruous with the results; stating that “Based on this frequency of detection, universal germline screening may be considered in all patients with CRC” when this study found lower frequency than previous studies is slightly mis-leading. Please consider removing or modifying this statement.
Response 12: We have revised the final sentence to state, “Despite lower incidence of PGVs in this population compared to prior reports, this incidence of detection would still suggest that universal germline screening may be indicated for all patients with CRC.”
Reviewer 2 Report
Comments and Suggestions for Authors
This is a very well written manuscript describing the genomic landscape of Early-Onset (EO) versus Average-Onset (AO) colorectal cancer populations (n=11006) in tumor tissue. In addition 6311 patients underwent germline mutation analysis. The size of the panels used was limited, but alterations analyzed were very well chosen according to the relevance for colorectal cancer and the large sample size is a strength of the research project. The methods applied are sound and the data are presented clearly. The discussion covers current knowledge on EO colorectal cancer and the findings of this study are discussed very well in this context.
Author Response
This is a very well written manuscript describing the genomic landscape of Early-Onset (EO) versus Average-Onset (AO) colorectal cancer populations (n=11006) in tumor tissue. In addition 6311 patients underwent germline mutation analysis. The size of the panels used was limited, but alterations analyzed were very well chosen according to the relevance for colorectal cancer and the large sample size is a strength of the research project. The methods applied are sound and the data are presented clearly. The discussion covers current knowledge on EO colorectal cancer and the findings of this study are discussed very well in this context.
Response: We appreciate this favorable feedback.
Reviewer 3 Report
Comments and Suggestions for Authors
The article "Genomic Landscapes of Early-Onset versus Average-Onset Colorectal Cancer Populations" by Storandt and colleagues is an interesting study on colorectal cancer, using several tools and expertise ranging from clinical oncology to bioinformatics. While short, the paper goes straight to the point of checking differences between early and middle onset CRC, and I appreciated its focus and pragmatism. I have however a few points to raise.
- In the brevity of their study, the authors do not perform what is in my opinion a mandatory step in this case: co-occurrence analysis. Are some of the detected somatic and germline mutations co-occurring? Are these co-occurrences different for eoCRC and aoCRC groups? For example KRAS and TP53 are co-occurring in some cancer types, following the vague dogma of cancer needing mutations in both an oncogene and a tumor-suppressor gene. In order to do this, even a simple Fisher's Exact test between all pairs of genes shown in Figures 1 and 2. Normally one would also correct co-occurrence with linkage information, but I don't think that's necessary here, because all genes seem far away from each other on the genome.
- Line 110: the eoCRC and aoCRC groups are a bit imbalanced, with the second group being 4 times as big as the first, and the criterion of separation may be too sharp. Would a different distinction provide a chance to find significant results? For example, the authors could separate the groups in <50 yo and >60 yo. Now 50 years old is a sharp separator between the two groups, which may be a problem if the actual differences happen in a different age (e.g. 55). Also, patients with close ages (49y11m old and 50y1m old) are forcibly separated, which seems too strict and arbitrary. I know eoCRC and aoCRC are defined using this threshold in literature, but we are not forced to follow definitions if they go against scientific pursuit.
- MINOR: Line 179: large numbers should preferably be shown as numeric values, so "One thousand four hundred and fifteen" should really be "1,415", especially because the number is compared with "4,898" later in the sentence.
Author Response
- In the brevity of their study, the authors do not perform what is in my opinion a mandatory step in this case: co-occurrence analysis. Are some of the detected somatic and germline mutations co-occurring? Are these co-occurrences different for eoCRC and aoCRC groups? For example KRAS and TP53 are co-occurring in some cancer types, following the vague dogma of cancer needing mutations in both an oncogene and a tumor-suppressor gene. In order to do this, even a simple Fisher's Exact test between all pairs of genes shown in Figures 1 and 2. Normally one would also correct co-occurrence with linkage information, but I don't think that's necessary here, because all genes seem far away from each other on the genome.
Response: We have completed a co-occurrence analysis now, but this study is likely overpowered to assess as there are over 300 significant co-occurring mutations among those with eoCRC alone, and the volume of this data likely lacks clinic significance. We can show co-occurrence through Supplemental Figures 4 and 5, which show relatively similar co-occurrence profiles between those with eoCRC and aoCRC. We now make mention of this is the results section, to draw the reader’s attention to this.
- Line 110: the eoCRC and aoCRC groups are a bit imbalanced, with the second group being 4 times as big as the first, and the criterion of separation may be too sharp. Would a different distinction provide a chance to find significant results? For example, the authors could separate the groups in <50 yo and >60 yo. Now 50 years old is a sharp separator between the two groups, which may be a problem if the actual differences happen in a different age (e.g. 55). Also, patients with close ages (49y11m old and 50y1m old) are forcibly separated, which seems too strict and arbitrary. I know eoCRC and aoCRC are defined using this threshold in literature, but we are not forced to follow definitions if they go against scientific pursuit.
Response: This is a well taken point. We agree that the distinction between eoCRC and aoCRC is entirely arbitrary. While this analysis would be valuable, we are unable to complete this analysis at this time. Our hope is that our current data will contribute to our current understanding of genetic profiles among patients with eoCRC and aoCRC, and by using previously established definitions, allow for comparison to prior studies.
- MINOR: Line 179: large numbers should preferably be shown as numeric values, so "One thousand four hundred and fifteen" should really be "1,415", especially because the number is compared with "4,898" later in the sentence.
Response: While this is generally true, when a number begins a sentence, it is considered grammatically proper to use long form, as opposed to a numeric value.
Round 2
Reviewer 3 Report
Comments and Suggestions for Authors
The authors have improved their manuscript to a very acceptable level.